# Robots and Robotics in Nursing

**DOI:** 10.3390/healthcare10081571

**Published:** 2022-08-18

**Authors:** Gil P. Soriano, Yuko Yasuhara, Hirokazu Ito, Kazuyuki Matsumoto, Kyoko Osaka, Yoshihiro Kai, Rozzano Locsin, Savina Schoenhofer, Tetsuya Tanioka

**Affiliations:** 1Department of Nursing, College of Allied Health, National University, Manila 1008, Philippines; 2Graduate School of Health Sciences, Tokushima University, Tokushima 770-8509, Japan; 3Department of Nursing, Institute of Biomedical Sciences, Tokushima University, Tokushima 770-8509, Japan; 4Graduate School of Sciences and Technology for Innovation, Tokushima University, Tokushima 770-8506, Japan; 5Department of Psychiatric Nursing, Nursing Course of Kochi Medical School, Kochi University, Kochi 783-8505, Japan; 6Department of Mechanical System Engineering, Tokai University, Hiratsuka 259-1292, Japan; 7Christine E. Lynn College of Nursing, Florida Atlantic University, Boca Raton, FL 33431, USA; 8Independent Researcher at Nursing As Caring, Jackson, MS 39202, USA

**Keywords:** artificial intelligence, robots, robotics in nursing

## Abstract

Technological advancements have led to the use of robots as prospective partners to complement understaffing and deliver effective care to patients. This article discusses relevant concepts on robots from the perspective of nursing theories and robotics in nursing and examines the distinctions between human beings and healthcare robots as partners and robot development examples and challenges. Robotics in nursing is an interdisciplinary discipline that studies methodologies, technologies, and ethics for developing robots that support and collaborate with physicians, nurses, and other healthcare workers in practice. Robotics in nursing is geared toward learning the knowledge of robots for better nursing care, and for this purpose, it is also to propose the necessary robots and develop them in collaboration with engineers. Two points were highlighted regarding the use of robots in health care practice: issues of replacing humans because of human resource understaffing and concerns about robot capabilities to engage in nursing practice grounded in caring science. This article stresses that technology and artificial intelligence are useful and practical for patients. However, further research is required that considers what robotics in nursing means and the use of robotics in nursing.

## 1. Introduction

According to the World Health Organization (WHO) [1], between 2015 and 2050, the percentage of the global population aged 60 years old and above will nearly double, from 12% to 22%. This shows that the aging population is increasing at a rate considerably greater than in the past. Thus, many countries face significant issues concerning the healthy living of older persons, ensuring that health and social systems are prepared to take advantage of this demographic shift. For this reason, some countries have developed the integration of technologies capable of human interaction, such as robots with artificial intelligence (AI) [2]. These technologies are particularly useful in hospital settings, in which demands for healthcare, in general, can result in a shortage of healthcare workers [3].

When the system application involves sophisticated technologies such as robotics in nursing, Frazier et al. [4] declared that with nurses constituting 45% of all healthcare professionals in healthcare practice, understaffing continues to be evident as a priority problem today. Thus, the application and deployment of complex technologies as systems of care, such as healthcare robots, are becoming more important [5]. The use of robots may also potentially provide enhanced patient outcomes due mainly to the usage of technologies in healthcare. The aging population demands competencies with technologies as crucial to attaining, maintaining, and sustaining human health and well-being [6]. The expected outcomes involve increased efficiency and provide supporting and supplementing of understaffing [7]. With nursing practice grounded in caring science paving the way towards transcending dependency with technologies [8], healthcare workers are made increasingly aware of the anticipated outcomes of technological dependency, exacerbated by the emergence of a pandemic [9].

Advancements in technology have led to robot development for nursing practice as potential partners to supplement understaffing and to provide efficient health care to persons with disabilities, older persons, or vulnerable persons [10,11,12]. Since the use of technologies helps and assists with procedures such as surgery, these technologies have been improved and are being used in other aspects of healthcare, from treatments to rehabilitation care. The potential use of robots in nursing and other health care disciplines could be for improving the accuracy and speed of the detection of illnesses to improving end-of-life care by helping persons maintain their independence for a longer period of time.

As the development of technology progresses, anticipated expressions of complex processes are expected. Becoming a progressively complex system is evidenced in the management of health care practices, especially in the strategies invoked in the usage of management systems [13].

Our insights are informed by nursing theoretical perspectives such as Boykin and Schoenhofer’s theory of Nursing As Caring [14], Locsin’s theory of Technological Competency as an Expression of Caring in Nursing and Healthcare [15], and Tanioka’s Transactive Relationship Theory of Nursing [16], as well as extensive research and analysis.

This article aims to discuss relevant concepts on robots and robotics in nursing and examine the distinctions between human beings and healthcare robots as partners and robot development examples and challenges.

## 2. Robots and Robotics in Nursing

### 2.1. What Are Robots in Nursing?

Due to the progressive increase in the aging population globally, estimated at 703 million aged 65 years old and above, the demands for older person care have also been heightened [17,18]. For this reason, an opportunity is being realized for the design and development of healthcare robots in nursing, fitted with artificial super intelligence (ASI) and capable of delivering nursing interventions and menial tasks in hospital settings. To bestow care for patients, specifically aging patients [19], with efficiency and accuracy, robots in nursing were defined by von Gerich et al. [20] following the definition of the International Organization for Standardization 8373 as “systems of mechanical, electrical, and control mechanisms used by trained operators in a professional health care setting that perform tasks in direct interaction with patients, nurses, doctors, and other health care professionals and which can modify their behavior based on what they sense in their environment”.

Similarly, Christoforou et al. [10] have also indicated that nursing robots can “serve as supplemental healthcare workers in hospitals, older-person care facilities, and at home. Robots in nursing can perform logistical and laborious physical tasks, combat loneliness and inactivity in the older population, or can be assigned to routine tasks such as measuring patients’ vital signs”. Additionally, other hospital technologies can also be integrated with robotic technologies, such as electronic health record systems that facilitate the recording of a patient’s healthcare history to ensure the continuity of care [8]. This assistive robotics can promote patient and nurse communication as care time may be enhanced by such robotic technology [10].

Meanwhile, according to Frazier et al. [4], robots are often programmed with sophisticated sensors, transmitters, and receivers as essential components. Moreover, robot computer systems are integrated with a display device, which has data sensing programs generated through the patient’s condition. A robot functions in such a way that when it senses (recognizes) patient physiological conditions in specific situations, the transmitter will communicate the data to the display device and be stored in the patient database. This healthcare robot functions through an ASI, which has different systems and sensors and delivers nursing care services efficiently [21]. These efficient and accurate functionalities have become the priority operational functions desired for developing robots in nursing [19,21].

### 2.2. Robotics in Nursing

The inclusion of advancements in technologies is heightened by the demands of healthcare in a highly technological world. The preference for a reimagined landscape fully determined by a humanizing care environment, as emphasized in Locsin’s theory in which technology, caring, and nursing have become inseparable, coexisting as conceptual models of humanizing care [21].

Robotics [22] is the engineering and operation of machines that can autonomously or semi-autonomously perform physical tasks on behalf of a human. Typically, robots perform tasks that are either highly repetitive or too dangerous for humans to conduct safely. Mechanical robots use sensors, actuators, and data processing to interact with the physical world. Someone who makes their living in robotics must have a strong background in mechanical engineering, electrical engineering, and computer programming. Recently, the field of robotics has begun to overlap with machine learning and AI.

Robotics in nursing practice continues to be a challenge to ethical deployment to ensure safe, secure, competent, and emotive functions of healthcare robots. Pivotal to the recognition of robots as partners in nursing is their continued proficiency, continuously deliberated with future policies and regulations in mind [23]. As stated by Maalouf et al. [24], distinct functional foci are represented by various types of healthcare robot applications in the diversified subject of robotics in nursing. Assistive robots and socially assistive robots were the main categories. The field of robotics in nursing is evolving fast to cope with the need for help in caregiving, especially for the elderly and individuals with disabilities. The future development of robotics in nursing depends on a series of improvements in theory and applications.

In ISO 8373:2021 (en) Robotics–Vocabulary [25], Robot is defined as follows: “Robot is a programmed actuated mechanism with a degree of autonomy to perform locomotion, manipulation, or positioning (3.1)”. In other words, things such as devices for disease management and symptom control of patients, and robotics houses in which the building itself is robotized, are all included in the robots.

However, this definition will change (rather actively) according to the times, and even if something is not strictly according to the ISO definition now, it could become a robot in the future if many people start calling it a robot. The research field of robotics in nursing deals with robotics to improve the quality of nursing care, beginning with the use of technology, a major concept to enable nurses to provide beneficial care to nursing subjects.

For nurses, incorporating robotics into nursing means working to improve the quality of nursing care and reduce workload. For patients, the robot can be effective in maintaining or treating their healthcare needs or improving their QOL or physical functions. Robotics in Nursing is an interdisciplinary discipline that studies the methodology, technology, and ethics for developing and using robots that support and collaborate with nurses in the nursing field. Robotics in Nursing is an interdisciplinary discipline that studies methodologies, technologies, and ethics for developing robots that support and collaborate with physicians, nurses, and other healthcare workers in practice.

Robotics in nursing is geared toward learning the knowledge of robots for better nursing care (including safety, functions, and effects of robots, and how to use them), and for this purpose, it is also to propose the necessary robots and develop them in collaboration with engineers. However, nurses are not typically educated to understand all systems and machines using mathematics and physics as engineers do. Robotics in nursing aims to help nurses use robotics to provide the latest and most effective care to nursing patients by having an affinity for engineering and always working closely with engineers and engineering researchers.

When introducing a developed robot, it is important to create an environment for the effective use of that robot. Specifically, it is necessary to clarify how the robot should function as a team member with caregivers who connect the robot and patients, such as doctors and nurses, evaluate the results of better nursing care by patients and nurses, and provide feedback for developing new robots. This is necessary to better robot-assisted nursing care.

Nursing researchers who play the role of advocating the rights of robot users should not only promote development but also sometimes decide to stop or change the way the robots are used. This includes knowledge of content related to AI technologies that control the autonomous actions, statements, and decisions of robots to communicate with patients and nurses.

In robotics in healthcare, it is important to think of the three-party relationship between patients, nurses (healthcare professionals), and robots and to effectively use robots and AI as tools and technologies in this relationship. In this context, it is important to effectively use robots and AI as tools and technologies:To study the knowledge of robots (including safety, functions, and effects of robots and methods of use) for better nursing care.To propose robots necessary for better nursing care and to develop such robots in cooperation with engineers.To develop an environment for effective use of robots when robots are introduced for better nursing care.To examine the use of robots and AI from ethical and moral viewpoints.To model how doctors, nurses, other healthcare workers, and robots should function as a team for better care.To evaluate the results of better care by patients, doctors, and nurses, and to provide feedback for developing new and improved robots.

## 3. Distinctions between Humans and Robots

### 3.1. Nursing Viewpoint

In the nursing setting, Rogers [26] stated that human beings are pandimensional energy fields that cannot be reduced to parts or divided, in which patterns manifest to have specific characteristics of being whole. Therefore, they are unpredictable beings from the knowledge of the parts. Fawcett and De Santo-Madeya [27] supported this definition in that person cannot be predicted by viewing, labeling, or summarizing them as they are considered unitary wholes with unique features. Furthermore, attributes of persons as caring can contribute to the possibility that robots can manifest caring [25]. Boykin and Schoenhofer [14] have expressed that people are caring by virtue of their humanness. As the manifestation of caring is substantiated by philosophical, theoretical, and theological views, the manifestation may be appropriated as caring.

### 3.2. Behavior of Robots-Like Humans

Contemporary healthcare robots will require robotics engineering programming in order to simulate caring practices, which can eventually be likened to human caring practices. Concerns about theocratic and philosophic perspectives remain critical determinants in assuming the possible similarities between humanoid robots and human beings’ expression of “caring” practices, particularly reflective of caring attributes.

Robots can be programmed by humans to act as human-like as it is currently possible to attend persons’ healthcare. While the possibility of intended acts by humanoid robots is to engender activities much like human nurses, this possibility depends on the technological advances of the times. In collaboration with persons expressing human caring practice, robots can or may be able to render functions that can simulate those of the human nurse. Locsin et al. [28] explained that caring in nursing refers to the relationship that exists between nurses and the persons being cared for. Caring encompasses empathy for and connection with people [29]. Nonetheless, if one refers to the theological and philosophical concepts of the humanness of persons as primary determinants of humanness, then robots have yet to be identified as equally the same as human beings; otherwise, humanoid robots can simply be functional technologies—instruments that facilitate human nurses’ expressions of caring. However, based on humanoid robot acceptance by humans in healthcare, the development of humanoid robots to express human caring as an “autonomous” being may create a heightened awareness of the issue of sentience, creating additional concerns that may help, or not, in advancing human science and human care in a world dense with technologies.

## 4. Physical Attributes and the Aspect of Expressing Humanness

### 4.1. Physical Aspects and Characteristics That Express Humanness

Understanding the differences between humans and robots in terms of their capability to care can be divided into two aspects: physical aspects and the aspect of expressing humanness. Today, the physical differences between humans and robots are obvious yet may not be as distinctive in the future. Current advances in researching human skin-like prostheses have become popular [30,31,32]. As stated by Liu et al. [32], tactile sensing plays a vital role that will develop the cognition and intelligence of robots as it becomes easier for the robots to explore their surroundings autonomously. For example, humans deprived of reliable tactile information, such as being numb or having cold fingers, may become clumsy and can create accidents. Robotic systems should have the feature of being able to sense touch to safely interact in uncertain environments, such as offering care for human beings.

It is said that human nurses are considered resources for health, yet there is a crisis in their numbers due to human aging and vulnerability to diseases as organic beings. It is inevitable that human resources would be scarce because of limitations of life expectancy and the physical capabilities of human beings as organic beings. Furthermore, decline of human births with the increasing rate of the aging population and developed medical technology and/or complex medical systems altogether bring concerning situations for the future availability of human resources for healthcare [33]. Corollary to these concerns of human resource allocation is the proximity of human beings to diseases and illnesses. The American Nurses Association [34] has recognized these limitations, adding personal risks of harming nurses. Robots and robotics in nursing settings would be able to address the resource problems because robots are mechanical and inorganic, and therefore, robots do not succumb to diseases and other consequences of being organic. As sophisticated technologies as these are, robots have their inherent problems such as sensory fidelity, the potential for being “hacked”, and mechanical deterioration, thus potential inability to ensure safe, secure, and precise activities of healthcare practice.

The advantages of having robots in healthcare facilities have produced the idea that they may outperform humans in some tasks, and replacing human nurses with robots could be a welcome possibility [19]. However, issues concerning robot capabilities of expressing human-like compassion [35] and empathy [25] are contemporary discourse topics. van Wynsberghe [36] also stated that healthcare robots do not currently have the competencies to express “caring in nursing” that is expected of a human nurse. Other critical topics regarding robot sentience [37] are popular themes of discourse as well.

### 4.2. Argument on whether Robots Are Capable of Having a “Soul”

Topics on the capability of robots to experience feelings and sensations refer to the argument that the current generation of humanoid and android robots already have the look and behavior that allows the robots to be accepted by people as peers. Another critiqued view is the acceptance as a form of deception in which robots pretend to be sentient through their behavior [38]. However, the argument on whether robots are capable of having a “soul”, arguing that the soul allows human beings to be able to manifest caring and “be with” persons, will introduce various concepts relating to the human soul. That correlates with beliefs about God and how the idea of the humans’ unique capability of caring is endowed by a god as the human soul is said to be reflected in a god’s being [39].

From an Islamic perspective, human beings are the best creation of God, and from a Judeo-Christian viewpoint, a person or a human is created with God’s own image, including his ability to be good [40]. However, human beings have no permanent existence and are always changing, according to a Buddhist perspective. Though in Shinto, which is a Japanese god spirit that is a good being, people become “kami” or God-like after dying. Despite the concepts allowing for evolving explicatory discourse on Nursing, the topics are still a delicate issue to be questioned when there is an alteration to the wholeness of a person to attain something beyond the limitations of a human [26]. Some philosophical and theoretical views of persons, such as the oneness and co-existence of the soul and body, are asserted as material to the understanding of human beings as persons.

Wilson [41] confers individuality and humanity to humans, emphasizing the duality of the mind and body. This means that it is not necessarily the soul that is the essential component of being human. Additionally, there are popular discussions on altering the wholeness of the person, creating a debate about whether it affects the beliefs about the human being as a perfect image of God. Terms such as Transhumanism and Posthumanism give the idea that the wholeness of a human can still go beyond its limits and evolve while it is still part of its growth to perfection. Transhumanism is the concept of human evolution that will lead to us being better humans by combining technology and biology [42]. Posthumanism refers to the use of AI, which allows evolution in humans genetically or bionically [43]. Such concepts are argued to be the final stage for humans to reach transcendence as a being. This further creates the idea of how far and complex it would take to conclude what allows a human to care as such, just by multiple concepts and issues.

For the question of whether robots can have a soul, they do not, as the concept of a soul focuses on the theological basis of the philosophical truth of the existence of a soul, while having the “mind” is another neuro-physiological and biological concept [44].

### 4.3. Robots May Communicate in Human-Like Manner

Robots may be able to communicate with their beings in a human-like manner, mimicking persons through natural language processing. Ren and Matsumoto [45] and Wagoner and Matson [46] support this probability, as they have also explained the capabilities of the process in their studies. Moreover, there have already been instances where robots mimicking animal-like behavior and communication have been achieved. Zhakypov et al. [47] designed small robots that were programmed to emulate how ants worked, from their structure within the colonies and how they work with one another through the usage of communication. This development can eventually progress into human-like behavior. This was similar to studies such as Nishio et al. [48], which explored the effectiveness of prolonged conversations with older persons through twin robots, finding that it was effective for more than half of the participants. These robots can emulate the behavior and even social hierarchies to a certain degree. Through this, they may have the capabilities of transcending from emulation of human-like manners to that of a simulation between two (2) robots.

## 5. Robot Development Examples and Challenges

Japan has been implementing the “Japanese Robot Strategy”, a program that encompasses the use of communication robots for older persons who have low levels of social participation. This strategy is pivotal in preventing dementia and delaying chronicity while promoting positive therapeutic effects [12]. An illustration of this strategy is the use of service robots. One popular robot used in industry, entertainment, and patient care is the humanoid robot called Pepper. It is a service robot programmed to engage persons who have certain conditions in therapeutic activities, such as monitoring their performance of tasks that help improve their physical functions as well as the extent of their interactive communication [49]. A similar robot is the Telenoid, a robot teleoperated with the intent to communicate with older persons who are diagnosed with Alzheimer’s disease [50].

Taking the situation during the COVID-19 pandemic as an example, several robotics companies developed their robots to contribute to the medical field. As mentioned by EHL (Ecole hôtelière de Lausanne) Faculty [51], companies such as UVD (Ultraviolet Disinfection) robots, PAL Robots [52], and TIAGo (Take It And Go) had their robots used in hospitals to perform various tasks that would otherwise place staff in danger due to their exposure to the virus; functions such as disinfection of areas by UVD robots in Italian hospitals, PAL’s ‘Ari’ robots to increase social stimulation, and TIAGo’s dual purpose robots of disinfecting and delivering medicine and food to patients. What was more interesting was the mention of Boston Dynamics’ work in developing a robot, their dog-like robot called ‘Spot’ that could take vital signs from a distance of two (2) meters from the patient by developing algorithms that can aid in such measurements [53]. Although the latter example is still being worked upon, the potential of robots in aiding healthcare professionals, especially nurses, can come to fruition once the technology becomes available to allow these concepts/plans to be part of the field.

### 5.1. Experiments Exploring Robots and Their Interactions with Humans

One study by Betriana et al. [54] explored the characteristics of the interactive communication between Pepper, patients with schizophrenia, and healthy persons. What transpired were exercises performed for both human groups where similarities and differences in their experiences were identified. The study results showed that Pepper the robot could successfully provide effective communication for both patients with mental health conditions and those who were healthy. However, due to the nature of how Pepper was operated (remotely), certain characteristics such as the response time during conversations, gaze, and entertainment functions would need to be improved.

Another study by Yamazaki et al. [55] explored the use of AHOBO, a frail care robot, to assist older adults living at home. In the study, two support systems were used, blood pressure measurement for the physical aspect and reminiscent coloring as a recreational activity for the psychological aspect. Based on the subjective evaluation, the results confirmed that the suggested robot has no effect on blood pressure readings and is satisfactory in terms of simplicity of use. Subjective evaluation of the reminiscent interaction was conducted on two older adults using the verbal fluency task, and it was confirmed that the interaction can be employed in daily life. The use of robots that currently exist in the field, such as Pepper and AHOBO, are still being improved upon, as the technology is still in its early stages. Therefore, the potential improvements that they can realize, especially with the addition of AI, can greatly improve their functions and be used in the realm of nursing. The potential of robots and AIs in the field of nursing and medicine can be endless, especially with the technological advancements that have been made in the past century that have opened up to devices such as self-driving cars, facial recognition, and many more.

### 5.2. Robots with Natural Language Expression by AI

The viewpoints on robots and AI are discussed to enable nurses to work more effectively and securely while providing patient care [56], but in doing so, robots with AI must first discover nursing situations occurring based on data previously stored in an envisioned “Nursing Situation Database”. The appropriate nursing response must be recognized from the “Nursing Response Database”. For example, if a patient has a hallucination, the AI will find a nursing response for that patient from the database, and the nursing robot will respond appropriately (appropriate actions and natural language expressions). However, not all nursing situations are stored in the database. Therefore, it is necessary to collect big data on nursing situations and nursing responses that the nursing robots have actually faced. A mechanism is needed for the “nursing situation database and its response database” to self-learn and evolve. This, in turn, would require nurses to be trained to work alongside the technology to provide the best treatment and experience for the patients.

Most studies involving robots and AIs would concern themselves with assistive functions in general, though there have also been studies that explore the potential nursing robots (as well as other robots) could have in the near future. According to Christoforou et al. [10], the functions of nursing robots can serve as supplemental healthcare workers, whether at home or in hospitals, as they can be assigned to perform the logistical and laborious aspects of nursing, whereas remote-controlled telerobots can take over the interactive caretaker duties. It should be taken note that even in this study, the functions of nursing robots are limited to that of their human counterpart’s guidance when it comes to interactions with patients. These robots can be further used when there are cases where interacting with patients could be dangerous for the staff, such as outbreaks. For instance, Yamazaki et al. [57] studied the use of Telenoid (teleoperated robots) in a residential care facility to examine how the older adults with dementia reacted to it. The findings revealed that an affectionate bond may grow between the older adults and the android, allowing the operator to communicate easily with older individuals and elicit answers.

With the advancement of AI technology, natural language interaction with patients can become a reality. However, because natural language dialogue requires advanced processing, many issues exist that need to be resolved. For example, there is the problem of omission. When people interact with robots using natural language, they sometimes omit important words that impact the comprehension of the sentence, such as subjects and objects. Matching analysis technology is indispensable. Ambiguous expressions without antecedents due to poor memory or poor judgment are also a problem in understanding meaning and intention.

Approaches to deepening understanding by having a robot repeatedly ask questions in response to unintended utterances [58] and methods to perform moral reasoning to resolve the ambiguity without asking questions [59] have also been studied. However, it is difficult for robots to make judgments that humans can make from context. Alternatively, an accurate understanding of intentions may not necessarily be necessary when miscommunication in dialog does not have serious consequences.

It is not difficult to imagine that an AI-powered robot that understands human emotions and can engage in chit-chat would not only reduce the burden of caregiving and nursing but would also be friendly and have a positive impact on patient health. In caregiving and nursing chats with patients, emotional understanding is more important than an accurate understanding of the intent of the speech. It is also possible to understand the health status of the patient through chatting [60].

## 6. Conclusions

Technology and AI are useful and practical for patients. Robotics in nursing is an interdisciplinary discipline that studies methodologies, technologies, and ethics for developing robots that support and collaborate with physicians, nurses, and other healthcare workers in practice. Robotics in nursing is geared toward learning the knowledge of robots for better nursing care, and for this purpose, it is also to propose the necessary robots and develop them in collaboration with engineers. However, further research is required that considers what robotics in nursing means and the use of robotics in nursing. There is still a lack of study on whether they are capable of replacing humans due to human nurses’ ability to manifest caring relates to their humanness or their unpredictable nature. One of the most important, in our opinion, would be to work on the Nursing Situation and Response Databases. The empathic capacities that robotics and AI can demonstrate for humans can exist through programmed activities. The knowledge generated will bring information to engage in relationships between empathy and AI and contribute to understanding its usefulness and impacting nursing/caring theories.

## Data Availability

The data presented in this study are available on request from the corresponding author. The data are not publicly available due to privacy and ethical restrictions.

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
