# Peer review of "Robots and Robotics in Nursing"

_healthcare, 2022, doi:10.3390/healthcare10081571_

Round 1

Reviewer 1 Report

Technological advancements have led to the use of robots as prospective partners to complement understaffing and deliver effective care to patients.

The authors state their own opinion on robots from the perspective of nursing theories and robotics in nursing and discusses the distinctions between human beings and healthcare robots as partners and robot development examples and challenges based on a literature review.

Robotics in nursing is an interdisciplinary discipline that studies methodologies, technologies, and ethics for developing robots that support and collaborate with physicians, nurses, and other healthcare workers in practice. Robotics in nursing is geared toward learning the knowledge of robots for better nursing care, and for this purpose, it is also to propose the necessary robots and develop them in collaboration with engineers.

Authors highlighted two points regarding the use of robots in health care practice: (1) issues of replacing humans because of human resource understaffing. (2) Concerns about robot capabilities to engage in nursing practice grounded in caring science.

I agree with the authors concluding that: (a) the technology and artificial intelligence are useful and practical for patients. (b) However, further research is required that considers what robotics in nursing means and the use of robotics in nursing.

This is an interesting piece.

Some minor suggestions for the authors to improve the presentation.

1.  Insert a clear purpose. In the intro there is an opinion

2. insert a short method section explaining the design

3. Section 2-6 are fine and interesting. My suggestion is  to try to rearrange them into a section of results and a section of discussion

Reviewer 2 Report

The abstract states that “This article states our own opinion on robots from the perspective of nursing theories and robotics in nursing and discusses the distinctions between human beings and healthcare robots as partners and robot development examples and challenges based on a literature review.However, in the text of the article in each paragraph there are references to the literature and it is not clear where, in fact, the opinion of the authors is presented. At the same time, this article is not a review. In addition, the results of any scientific research or comparative analysis are not presented in this article either. Section 5 on artificial intelligence is written very superficially. It is not clear for what target audience this article is written. This article is about everything and nothing, it does not contain details, classifications and new provisions that are of interest to specialists. For them, the proposed material is quite obvious (banal) and is of no interest.

We can define the genre of this article as “popular science article” that may be of interest to ordinary people who do not conduct scientific research. If the journal's policy allows the publication of such articles, then it can be published without significant changes. Authors only need to check the references to the literature. For example, in lines 212, 229, 373, 381, etc., the numbers of references to the literature are indicated incorrectly.
